# Amorphous topological superconductivity in a Shiba glass

Kim Pöyhönen[1], Isac Sahlberg[1], Alex Weststrom[1] & Teemu Ojanen[1]

Topological states of matter support quantised nondissipative responses and exotic quantum particles that cannot be accessed in common materials. The exceptional properties and application potential of topological materials have triggered a large-scale search for new realisations. Breaking away from the popular trend focusing almost exclusively on crystalline symmetries, we introduce the Shiba glass as a platform for amorphous topological quantum matter. This system consists of an ensemble of randomly distributed magnetic atoms on a superconducting surface. We show that subgap Yu–Shiba–Rusinov states on the magnetic moments form a topological superconducting phase at critical density despite a complete absence of spatial order. Experimental signatures of the amorphous topological state can be obtained by scanning tunnelling microscopy measurements probing the topological edge mode. Our discovery demonstrates the physical feasibility of amorphous topological quantum matter, presenting a concrete route to fabricating new topological systems from non-topological materials with random dopants.

[1] Department of Applied Physics (LTL), Aalto University, P. O. Box 15100, FI-00076 AALTO, Finland. Correspondence and requests for materials should be addressed to T.O. (email: teemuo@boojum.hut.fi)

Topological states are characterised by integer-valued invariants[1, 2] that remain robust in the presence of imperfections. While topological properties can be studied independently of local order, spatial symmetries play a central role in virtually all material realisations. This is emphasised by the fact that the theoretical search for new topological materials extensively employs band structures and reciprocal space. While topological states are generically robust to disorder which breaks spatial symmetries, this is typically established by treating the disorder as an additional feature in a well-defined clean system. Even topological Anderson insulators[3, 4], where moderate disorder actually gives rise to nontrivial topological properties, rely crucially on a specific band structure of the clean system. The concept of disorder, almost by definition, implies the existence of an underlying ordered reference state.

The role of spatial symmetries in topological materials raises the question of how much spatial order is necessary for topological states to persist. In addition to the fundamental interest, possible realisations have far-reaching practical implications. The search for topological states has already moved beyond the elements found in nature to artificial man-made structures such as Majorana wires[5–7]. The wires have the advantage of potentially allowing topological computation, but to carry out this function they must be almost defect free, which illustrates a generic complication in top-down fabrication strategies. On the other hand, fabrication of topological matter with randomly distributed constituents, if possible, would avoid that obstacle and offer new opportunities. A recent discovery of a mechanical gyroscopic metamaterial[8], albeit a purely classical system, suggests that also amorphous topological quantum matter could be achievable. By studying the properties of long-range hopping toy models, Agarwala and Shenoy[9] pointed out the mathematical possibility of topological states with randomly localised states.

In this work, we propose the Shiba glass, depicted in Fig. 1, as a concrete physical realisation of amorphous topological quantum matter. Remarkably, we discover that (i) for a finite out-of-plane polarisation, the system supports topological superconductivity above a critical density, and despite complete absence of spatial order, (ii) the topological phase is extremely robust and protected by a mobility gap and (iii) the topological phase supports edge

modes, whose signatures can be observed in standard scanning tunnelling microscopy (STM) experiments. The Shiba glass is fundamentally different from disordered topological materials which rely on band structures and thus on the spatial order of clean systems.

## Results

**Theoretical description.** The studied amorphous topological superconductor is comprised of randomly distributed magnetic moments on a superconducting surface with a Rashba spin-orbit coupling. The moments can arise from magnetic atoms, molecules or nanoparticles. Regular 1D structures of this type have been predicted to host Majorana states[10–18] with supporting experimental evidence[19–21]. More recently, ferromagnetic 2D lattices have emerged as a promising platform for chiral superconductivity[22, 23] with a rich topological phase diagram[24,25]. Classical magnetic moments embedded in a gapped $s$-wave superconductor give rise to Yu–Shiba–Rusinov (YSR) subgap states[26], localised subgap states which decay algebraically for distances smaller than the superconducting coherence length. In 2D superconductors, such as layered systems, thin films and surfaces, the decay of the wavefunctions from the deep-lying impurity has a functional form $e^{-r/\xi}/\sqrt{k_F r}$, where $\xi$ and $k_F$ are the superconducting coherence length and the Fermi wave vector of the underlying bulk. The Shiba glass results from a hybridisation of randomly distributed YSR states. To model the system, we consider deep-lying YSR states with energies $\varepsilon_0$ located in the vicinity of the gap centre $\varepsilon_0/\Delta \ll 1$, where $\Delta$ is the pairing gap in the bulk. The energy of a single YSR state is given by $\varepsilon_0 = \Delta\frac{1-\alpha^2}{1+\alpha^2}$, where $\alpha = \pi J S \mathcal{N}$ is a dimensionless impurity strength, $J$ is the magnetic coupling, $S$ is the magnitude of the magnetic moment and $\mathcal{N}$ is the spin-averaged density of states at the Fermi level. The deep-impurity assumption translates to $|1 - \alpha| \ll 1$ and the energy of an impurity state is given by $\varepsilon_0 \approx \Delta(1 - \alpha)$. As outlined in the Methods section, the low-energy properties of the coupled impurity moments are modelled by a tight-binding Bogoliubov-de Gennes Hamiltonian[24]

$$H_{mn} = \begin{pmatrix} h_{mn} & \Delta_{mn} \\ (\Delta_{nm})^* & -h_{mn}^* \end{pmatrix}, \tag{1}$$

which describes a long-range hopping between YSR states centred at random positions $r_n$. The entries $h_{mn}, \Delta_{mn}$ for arbitrary configuration of magnetic moments is lengthy and given in Supplementary Note 1. Physical intuition can be obtained by considering the special case of fully out-of-plane ferromagnetic spins, where the model reduces to

$$h_{mn} = \begin{cases} \varepsilon_0 & m = n \\ \frac{\alpha\Delta}{2}\left[I_1^-(r_{mn}) + I_1^+(r_{mn})\right] & m \neq n \end{cases},$$

$$\Delta_{mn} = \begin{cases} 0 & m = n \\ \frac{\alpha\Delta}{2}\left[I_4^+(r_{mn}) - I_4^-(r_{mn})\right]\frac{x_{mn}-iy_{mn}}{r_{mn}} & m \neq n. \end{cases} \tag{2}$$

In the above expression $r_{mn} = |\mathbf{r}_m - \mathbf{r}_n|$, and $x_{mn}$ and $y_{mn}$ are components of $\mathbf{r}_m - \mathbf{r}_n \equiv (x_{mn}, y_{mn})$. The hopping elements are expressed in terms of the functions

$$I_4^\pm(r) = \frac{\mathcal{N}_\pm}{\mathcal{N}}\Re\left[iJ_1\left(k_F^\pm r + ir/\xi\right) + H_{-1}\left(k_F^\pm r + ir/\xi\right)\right],$$
$$I_1^\pm(r) = \frac{\mathcal{N}_\pm}{\mathcal{N}}\Re\left[J_0\left(k_F^\pm r + ir/\xi\right) + iH_0\left(k_F^\pm r + ir/\xi\right)\right], \tag{3}$$

where $J_n$ and $H_n$ are Bessel and Struve functions of order $n$. The Rashba spin-orbit coupling induces two helical Fermi surfaces with density of states $\mathcal{N}_\pm = \mathcal{N}\left(1 \mp \lambda/\sqrt{1+\lambda^2}\right)$ and Fermi

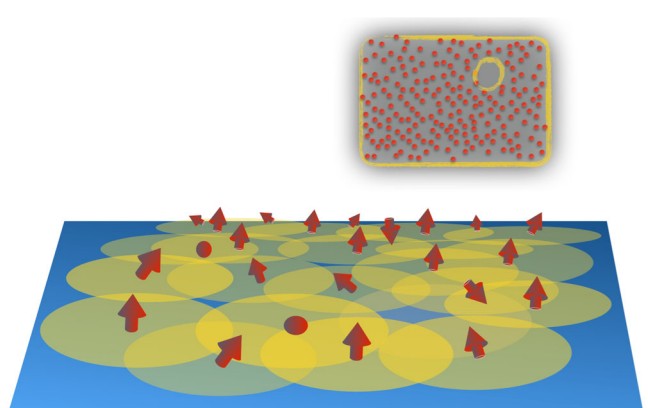

**Fig. 1** Structure of the Shiba glass. Magnetic moments (represented by a red arrow) on a superconductor bind a subgap Yu–Shiba–Rusinov state (represented by a yellow disc) centred on the moments. A Shiba glass results from a hybridisation of individual bound states in a random spatial distribution of moments. The collective amorphous state supports topological superconductivity above a critical moment density at finite out-of-plane polarisation. Inset: a finite sample is enclosed by a topological edge mode of chiral Majorana states. Rare fluctuations give rise to antipuddles that exhibit localised low-energy excitations within a mobility gap protecting the topological phase

wavenumber $k_F^\pm = k_F\left(\sqrt{1+\lambda^2} \mp \lambda\right)$, where $\lambda = \alpha_R/(\hbar v_F)$ is the dimensionless Rashba coupling and $k_F, v_F$ the Fermi wavenumber and velocity in the absence of spin-orbit coupling. The Rashba coupling also slightly modifies the superconducting coherence length $\xi = (\hbar v_F/\Delta)\sqrt{1+\lambda^2}$. For ferromagnetic textures, the pairing term $\Delta_{ij}$ vanishes with vanishing Rashba coupling $\alpha_R = 0$. The low-energy Hamiltonian (1) describes an odd-parity pairing $\Delta_{mn} = -\Delta_{nm}$, which is a long-range hopping variant of a $p_x + i p_y$ superconductivity. In Eq. (1) the hopping and pairing functions decay as $f(r) \propto \frac{e^{-r/\xi}}{r^{1/2}}$ and display oscillations at wave vectors $k_F^\pm$.

**Physical properties of the Shiba glass.** The spectrum and the topological phase diagram of a finite system can be calculated by diagonalising the effective Hamiltonian (1) for spatially uncorrelated random positions of magnetic moments. After deriving the finite-size properties, we discuss the extrapolation to the thermodynamic limit. For 2D time-reversal breaking topological superconductors, the relevant topological index classifying the state is the Chern number. We will evaluate Chern numbers by employing the real-space approach of Eq. (5).

By evaluating the Chern number, we uncover the topological phase diagram of finite Shiba glass systems which can be seen in Fig. 2a. For sufficiently high densities, a ferromagnetically ordered system is generally in a topological phase with Chern number

$|\mathcal{C}| = 1$. For the employed parameters, the critical density $\rho_c$ corresponds to the characteristic length scale $\bar{r}_c = \rho_c^{-1/2} \approx k_F^{-1}$. For lower densities $(\bar{r} \gg k_F^{-1})$, the system is in general topologically trivial and gapless; rare configurations can manage to enter a topological phase but do not survive disorder averaging. The pattern persists even when the directions of the local spins deviate from the perfect ferromagnetic configuration; in Fig. 2b we plot the phase diagram for spin configurations drawn from a thermal distribution where the angles $\theta_j$ between the moments and the surface normal are determined by the Boltzmann weights $e^{-\beta E_Z \cos\theta_j}$. This situation corresponds to an ensemble of decoupled spins at Zeeman field $E_Z$ polarising the moments perpendicular to the plane and disordered by thermal fluctuations at inverse temperature $\beta$. Alternatively, the situation can be regarded as a magnetic disorder where the disorder is parametrised by the thermal distribution and $\beta E_Z$ instead of some other random distribution. For $\beta E_Z = 10$, as indicated by Fig. 2b, the phase diagram remains qualitatively unchanged when compared to that for the completely polarised case. The robustness to moment disorder is not an artefact of the thermal distribution, and we discover qualitatively similar results for other disorder averages exhibiting comparable polarisation.

The physical consequences of the topological nature of the Shiba glass are illustrated in Fig. 2c, d. The first one shows that the local density of states (LDOS) is concentrated on the sample edges. This is a consequence of a topological edge mode enclosing

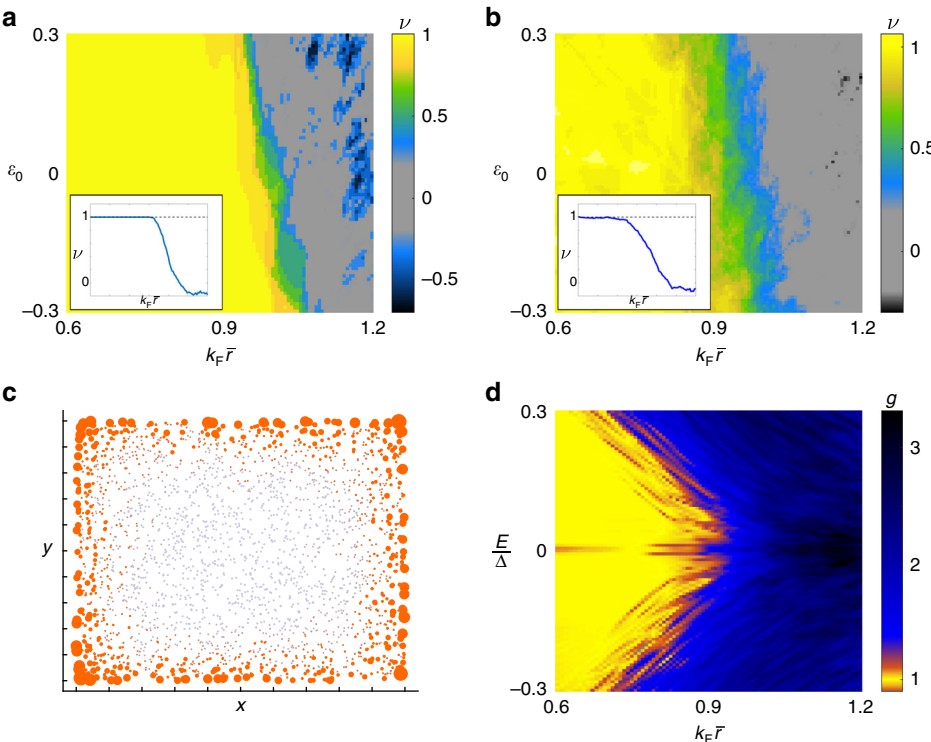

**Fig. 2** Topological superconductivity in the Shiba glass. **a** Topological phase diagram for a ferromagnetic Shiba glass as a function of the single-moment bound-state energy $\varepsilon_0$ and the characteristic length between the moments $\bar{r} = \rho^{-\frac{1}{2}}$, where $\rho$ is the moment density per unit area. The colour bar indicates the value of the Chern number. The adatom number is held fixed at 600, with $k_F\xi = \frac{4\pi}{5}$ and $\lambda = 0.2$. The displayed diagram is an average over 10 configurations. Inset: Line along $\varepsilon_0 = 0.1$, averaged over 500 configurations. **b** Same as in **a**, but for magnetic moment directions drawn from the Boltzmann distribution with $\beta E_Z = 10$ and averaged over 30 configurations, and with the number of moments fixed at 900. The deviation from the quantised values and the width of the transition region diminish as the system size is increased. **c** Local density of states (LDOS) for a $12.5\xi \times 12.5\xi$ square Shiba glass system comprising 2500 randomly distributed sites, integrated over subgap energies $|E| < 0.1\Delta$. Parameters used same as in **a**, with onsite energy $\varepsilon_0 = 0$. The areas of the orange discs correspond to the magnitude of the LDOS; each site is additionally represented by a grey point which is visible when the LDOS is negligible. **d** The thermal conductance (in units of $\frac{\pi k_B^2 T}{3\hbar}$) along the line $\varepsilon_0 = 0$ for the same system parameters as in the previous figures, but with 2500 adatoms. The vertical width of the conduction plateau (yellow) corresponds to the mobility gap of the system, and can be seen to close as the system approaches the transition to the trivial gapless phase

a finite system and is directly observable as discussed below. In Fig. 2d we have plotted the thermal conductance of finite systems coupled to external leads, as detailed in Supplementary Note 2. In the topological phase, the system exhibits a quantised thermal conductance which is a direct consequence of the nontrivial topology. The quantised conductance is effected by the edge modes despite the system being highly irregular in real space. In finite-size systems, for parameters close to the phase boundary, the quantised conductance plateau is destroyed and the conductance assumes continuous values. The non-quantised conductance in the trivial phase indicates that the low-energy states there extend over the sample.

The behaviour and exact phase transition point depends on the system parameters, though the overall trend of a topological phase at high densities remains. In Fig. 2 we have used parameters with high Rashba splitting $\lambda$ and low value of $k_F \xi$ as appropriate for a proximity-superconducting 2D semiconductor; a phase diagram for parameters more appropriate for metals are presented in Supplementary Fig. 2b, also indicating a transition to a topological phase at sufficiently high densities.

Now we turn to discuss the features seen when increasing the system size. First of all, in the thermodynamic limit the Shiba glass phase is gapless. While this is a generic feature of a superconductor with magnetic impurities[25], a qualitatively new mechanism for low-energy excitations arises in the topological phase. These emerge from rare fluctuations that leave a substantial area where magnetic moments are sparse. As depicted in Fig. 1, these empty antipuddles give rise to low-energy modes which are reminiscent of the gapless edge states circulating around a hole punched in a gapped topological phase. While the probability of formation of antipuddles is exponentially suppressed as a function of their size and their effect is relatively unimportant in finite systems with high density, in infinite systems antipuddles give rise to a tail down to zero energy in the DOS. The antipuddle mechanism provides a simple physical argument why the energy gap must scale to zero in the thermodynamic limit. The second important notion is that, in the thermodynamic limit, the system has well-defined topological nature despite being gapless. The low-energy modes, as we have argued above, are localised perturbations and the states with non-localised wavefunctions have a finite energy threshold. Thus, instead of an energy gap, the system exhibits a mobility gap protecting the topological state. This behaviour is analogous to the integer quantum Hall effect where the extended states carrying Chern numbers are separated by localised states in the Landau level gap[27]. In Supplementary Fig. 2b we have calculated the thermal conductance for an antipuddle configuration, which shows that for isolated antipuddles, the system has a vanishing energy gap but a finite well-defined mobility gap within which the heat conductance is quantised. In the topological phase the antipuddles are rare and effectively decoupled, thus they cannot destroy the conductance quantisation.

## Discussion
In our work, we have not addressed the question of the magnetic ordering, but rather show that a finite polarisation perpendicular to the plane gives rise to a topological phase. The nature of the ordering would likely depend sensitively on the specific physical realisation; however, there exist a number of mechanisms driving the system to a polarised state. For example, ignoring the modifications arising from superconductivity at large distances[28], the RKKY coupling leads to an effective interaction $H = \sum_{i \neq j} J\left(\left|\mathbf{r}_i - \mathbf{r}_j\right|\right)\mathbf{S}_i \cdot \mathbf{S}_j$ between the moments, where the sign of $J(r)$ oscillates as a function of the position as $\cos(2k_F r)$. While for distances $2k_F r \gg 1$ this leads to complicated frustrated

behaviour, in sufficiently dense systems $\pi/4 < k_F r < 3\pi/4$ the interaction is effectively ferromagnetic. Therefore, in the large part of the topological region $k_F \bar{r} \lesssim 1$, this mechanism favours a ferromagnetic ordering polarising the system. In addition, an anisotropic crystal field splitting $DS_z^2$ and an external Zeeman field $BS_Z$ would drive the system towards an out-of-plane polarisation.

The studied Shiba glass system could be realised by decorating an effective 2D or a layered 3D superconductor with magnetic atoms or molecules. Considering the requirement $k_F \bar{r} \lesssim 1$, dilute electron systems such as proximity-superconducting $2d$ semiconductors with Rashba spin-orbit coupling are promising candidate systems. Another candidate system is the layered superconductor NbSe$_2$ where $2d$ YSR states[29] and their coupling have been observed[30] recently. The most direct experimental probe is provided by STM measurement of the LDOS. As shown above, in the topological phase the Shiba glass system exhibits a significant concentration of the subgap LDOS at the sample boundaries, which can be directly observed by STM. This signal is clearly detectable at temperatures below the mobility gap scale which can be of the order of $k_B T = 0.1\Delta - 0.3\Delta$ as shown in Fig. 2d.

In summary, we introduced the Shiba glass as a platform for amorphous topological superconductivity and elucidated the general properties of such systems. Our results illustrate the physical feasibility of amorphous topological quantum materials and provide a concrete prescription to experimentally realise and observe them. Our discovery motivates expanding the search for topological materials beyond crystalline systems and paves the way for fabricating topological matter from nontopological materials with random dopants.

## Methods
**Real-space evaluation of the topological invariant**. To find the topological phase diagram, we need to evaluate topological invariants in real space. The relevant topological index for 2D systems with broken time-reversal symmetry is the Chern number. This is generally obtained in $\mathbf{k}$-space, but there are various methods of computing it in real space as well[31, 32]. A comparison shows that these methods are generally of similar computational efficiency and yield the same values for the topological invariant.

The real-space Chern number method of ref. [32] proceeds by defining the coupling matrices $C_{\alpha,\alpha+1}$, with elements

$$C_{\alpha,\alpha+1}^{mn} = \langle \psi^m | e^{i(\mathbf{q}_\alpha - \mathbf{q}_{\alpha+1}) \cdot \mathbf{R}} | \psi^n \rangle, \tag{4}$$

where $\mathbf{R}$ is the position operator, $\mathbf{q}_\alpha = \pi(\delta_{\alpha,1} + \delta_{\alpha,2}, \delta_{\alpha,2} + \delta_{\alpha,3})$ for $\alpha = 0, \ldots, 3$, and where $\psi^m$ are the eigenfunctions of the system with periodic boundary conditions. By use of these matrices, the Chern number is then obtained through the equation

$$\mathcal{C} = \frac{1}{2\pi} \sum_m \arg(\lambda_m), \tag{5}$$

with $\lambda_m$ being the complex eigenvalues of the matrix $C_{01}C_{12}C_{23}C_{30}$.

**Data availability**. Data sharing not applicable to this article as no datasets were generated or analysed during the current study.

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

## Acknowledgements

This work is supported by The Academy of Finland (T.O.), the Aalto Centre for Quantum Engineering (T.O.), the Swedish Cultural Foundation in Finland (K.P.) and the Vilho, Yrjö and Kalle Väisälä Foundation of the Finnish Academy of Science and Letters (A.W.).

## Author contributions

I.S. performed the numerical studies in the early stages of this work. K.P. and A.W. performed the analytical work and numerical calculations. T.O. planned and supervised the project. All authors analysed the results and discussed their interpretation. K.P., A.W. and T.O. prepared the manuscript.

## Additional information

**Competing interests:** The authors declare no competing interests.

