## [Peer Review File · Nature Communications]

Reviewers' comments:

Reviewer #1 (Remarks to the Author):

These authors proposed an amorphous topological superconducting phase in a Shiba glass. The system consists of an ensemble of randomly distributed magnetic atom on a superconducting surface. A magnetic impurity can induce the Yu-Shiba-Rusinov states within the superconducting gap, and the collection of the random magnetic momentum gives rise to the Shiba glass that form a topological superconducting phase. I think this is a new quantum phase of matter, a good example of "order out of disorder". It will be of interest to the community. If it can be observed experimentally, we shall have one more member of topological phases. The manuscript is well organised and written. I recommend it for publication.

Reviewer #2 (Remarks to the Author):

The manuscript "Amorphous topological superconductivity in a Shiba glass" is an interesting read and a curious result. While I have little doubt in the technical validity of the results, I would like to focus on context and wording of the statements in the manuscript. Those give me enough reason to recommend against publication of the article in Nature Communications.

1. Naming: Shiba glass should be explained. It is not spin glass, but I guess bears the name "glass" from the irregular bonds. This choice of naming should be explained.

2. Authors start from detailed discussion of why it is interesting to study topology in systems without underlying lattice as if it was a new concept (the citation in the section is from 2017). However, it is known since 2012 that aperiodic systems support the same topology as their periodic counterparts. See Phys. Rev. Lett. 109, 106402 (2012) and other works by the same authors as an example. Moreover, it is known that even weak invariants can be defined in aperiodic systems, see Rev. Lett. 116, 257002 (2016). Thus the novelty of the paper is not in proposing this concept of topology in aperiodic systems, rather proposing a realization. But, at the same time, the realization is also derived from the known one for the lattice counterpart. One of the authors has discussed a very similar system, albeit periodic, in Phys. Rev. B 93, 094521 (2016). This point constitutes enough reason for me to recommend against publication in Nature Communications and for sending to a more specialized journal.

3. Technical comments:

a) Colorscale is confusing. In fig. 2a,b the same color shows $\nu \sim 0.25$ and $\nu \sim -0.25$. In fig. 2d the same problem for above and below $g=1$.

b) It is unclear what is the purpose of "thermal" distribution of the spin magnetizations. Should not thermal case minimize the free energy of the spins together with the electrons on the impurities, not without them?

c) "conductance is effected by" -> "conductance is affected by"

d) The statement about STM measurement of edge states in a formally gapless system is confusing at first. The authors explain it later in the paper, but it would be illuminating to see an estimate of the edge state decay length as compared to typical size of the system where the rare configuration creates a near-zero (within T from zero energy) state.

e) In supplementary material it would be less confusing if the fig. 3b would have label "Thermal conductance" of "g" instead of only "Conductance"

f) Caption of supplementary Fig. 2a should state the size of the system.

Response to Reviewers' comments:

Response to the first referee:

We thank the referee for her/his efforts and are happy to learn about the enthusiastic reception of our work.

Response to the second referee:

We thank the referee for his/her efforts and the assessment that our work is technically sound. However, we strongly disagree with the assessment that the previous work on topological quasicrystals (such as PhysRevLett.116.257002) would in anyway undermine the novelty of our manuscript. Especially, the reasons and the previous works cited by the referee cannot form a logical basis for rejecting our manuscript for simple reasons.

First of all, we acknowledge in our paper that topological aperiodic systems have been studied before, actually for decades: all disordered topological insulators are aperiodic. The quasicrystals are very special aperiodic systems and forms a negligible subset in all nonperiodic systems. Also quasicrystals are spatially correlated, they even can even support the famous five-fold symmetric diffraction pattern. Therefore it is clear that they form a marginal subset in more general amorphous systems. We study a system with negligible spatial correlations which cannot be understood as a small generalization of quasicrystal order. In fact, from the point of view of spatial correlations quasicrystals may bare close resemblance to regular crystals.

Secondly, the crucial point of our paper is to introduce a concrete physical model where amorphous topological quantum state can be realized by *random doping*. This procedure could open flood gates in fabricating new realizations of topological quantum matter and devices. Therefore our work has general interest for the constantly expanding community of physicists working on quantum matter. It is clear that the previous quasicrystal work is irrelevant in this respect since fabricating quasicrystals resulting from random doping is likely impossible (actually it is easier even to fabricate perfect crystals with bottom-up techniques). Therefore quasicrystals are (at best) marginal in fabricating new solid-state quantum matter.

Finally, the fact that we have published previous work where topological states (for example PhysRevLett.114.236803) were obtained in regular magnetic lattices can no way be used as a reason to reject our current work. The magnetic adatom systems have already risen as competitors to semiconducting nanowires as a testbeds for topological superconductivity with enormous research activity. If anything, this makes our proposal even more interesting for general audiences since the experimental platform to study the amorphous magnetic systems is already under intense

study.

Response to the specific points:

a) Lattice systems with low or no spatial correlations are often called amorphous or “glassy” (even though they don't represent real glass), therefore an amorphous collection of magnetic Yu-Shiba-Rusinov impurities on a superconductor can be appropriately called as a “Shiba glass” in short.

b) As mention in the paper, one can regard the thermal distribution simply as a way to parametrize the disorder in spin direction. The disorder configuration was represented to point out that the specific ordering is not important as long as the state is polarized. But our parametrization could have a physical content as well. If a specific microscopic theory (possibly taking account complicated surface physics and chemistry as well as spatial randomness) predicts a ferromagnetic ordering with characteristic energy scale E_Z , then the studied disorder configuration corresponds to a thermal distribution at inverse temperature $\beta E_Z=10$. However, as discussed in the paper, the specific ordering can arise from interplay of various effects and we do not attempt to solve it from microscopies. For the existence of the topological phase it is sufficient that the state has finite perpendicular polarization.

c) The form “conductance is effected by” is also grammatically correct and more appropriate here. If the copy editor at later stages wishes, we will incorporate changes.

d) The fact that amorphous systems are gapless in the thermodynamic limit is also important point in our work and we are happy that the referee acknowledges this fact here. The low energy excitations arise from antipuddles that can be traced by STM similarly to the sample edges. For small systems with high density the probability of the antipuddles is so small that one does not expect to see them and the signal is what one would expect from a genuinely gapped state as illustrated by our Fig. 2 C. To estimate the typical edge penetration depth at high densities one can calculate what it is for a regular (say square) lattice with comparable density. It will be of the same order in the random lattice (in very dilute systems near the phase boundary this estimate becomes inaccurate since the edge states leak deeper into the system than in regular lattice).

e) We will implement the change of label suggested by the referee

f) The size of the system is not an appropriate measure in the figure since we calculate things as a function of density with the number of lattice sites fixed. The relevant information is obtained as a function of density and we want to spend equal computation time for each value of density and therefore fix the number of lattice sites. This means that the overall system size changes as a function of density.